# A One-Step MSE Estimation of Models in Production

## Abstract

In real-world operation of machine learning systems, monitoring the performance of prediction models is crucial. However, in these scenarios, actual values of target variables are observed with a delay, making real-time evaluation of prediction performance impossible. In this paper, we propose a novel one-step Mean Squared Error (MSE) estimation method that directly and tightly minimizes the upper bound of the MSE estimation error for regression tasks. Due to its direct estimation, our method is more efficient at estimating MSE compared to the conventional two-step approach, which approximates the mean and variance of the target variable. We also provide generalization error bounds for our proposed method based on a theoretical analysis. Our experiments demonstrate the effectiveness of our method, outperforming existing methods on both synthetic and real data sets.

## 1 Introduction

Machine learning (ML) has become widespread in real-world applications. To maximize the benefits of predictions made by machine learning models and minimize the damage caused by prediction errors, it is crucial to continuously monitor the predictive performance of these models (Kreuzberger et al., 2023; Ruf et al., 2021; Testi et al., 2022; Symeonidis et al., 2022). However, in real-world scenarios, the actual values of objective variables often experience a delay before observation (Plasse & Adams, 2016; Grzenda et al., 2020). Consequently, estimating the predictive performance of the model is essential for quality control in ML systems.

Estimation of prediction performance is commonly performed for classification tasks. Several works based on distribution shifts (Elsahar & Gallé, 2019; Do et al., 2021; Deng & Zheng, 2021; Schelter et al., 2020; Techapanurak & Okatani, 2021; Chen et al., 2021b) and "check" models, which are used to verify the correctness of predictions made by the prediction model, have been proposed (Chuang et al., 2020; Chen et al., 2021a). However, these methods pose challenges when applied to regression tasks because regression models deal with continuous target values, whereas classification models handle discrete target values.

One approach to estimating the Mean Squared Error (MSE), a prevalent evaluation metric for regression tasks, involves approximating the mean and standard deviation (or the confidence interval, the uncertainty) of the target variable ($y$) for each of input variables ($\boldsymbol{x}$). Subsequently, MSE can be estimated by averaging the squared differences between the approximated mean and the predictions made by the prediction model, plus the squared value of the approximated standard deviations (i.e., variances) for the evaluation inputs. Mean and variance estimation models (Nix & Weigend, 1994; Skafte et al., 2019) and uncertainty estimation methods (Lakshminarayanan et al., 2017; Wang et al., 2019; Liu et al., 2019; Rasmussen & Williams, 2005) are suitable for this MSE estimation approach. However, this approach requires learning two separate models for estimating the MSE: one for means and the other for standard deviations. Generally, we can compute the MSE of a prediction model with the mean and standard deviation for each input, but we cannot compute the mean and standard deviation using only the MSE or even the squared errors for each input. This implies that estimating means and standard deviations is more challenging than directly estimating the MSE. Following Vapnik's principle (Vapnik, 1998), which states that when solving a problem with limited information, one should not solve a more general problem than the original problem as an intermediate step, this two-step approach should be avoided due to its added complexity.

In this paper, we propose a novel one-step approach that directly estimates MSE based on the average of squared differences between the predictions made by the prediction model and those made by our "check" model. We derive an objective function that tightly bounds the MSE estimation error from above and train our check model by minimizing the objective. Our method effectively estimates MSE more efficiently than conventional two-step approaches due to its one-step direct estimation. Furthermore, we provide a theoretical analysis regarding the upper bound of our method's generalization error. We also propose a regularization term which aids in the learning of our check model. Experiments conducted using both synthetic and real-world benchmark data sets confirm that our method achieves the lowest MSE estimation error for all data sets.

Our key contributions are summarized as follows:

- We formulate the MSE estimation problem and propose a one-step estimation method that directly and tightly minimizes the MSE estimation error.
- We theoretically analyze our method and provide an upper bound of the generalization error for our MSE estimation approach.
- We conduct MSE estimation experiments using both synthetic and real-world data sets, empirically demonstrating the superior performance of our method compared to the conventional two-step approaches.

## 2 PRELIMINARY

In this section, we introduce the problem definition and relevant previous methods briefly.

### 2.1 PROBLEM FORMULATION

We consider a supervised regression problem; the input space is $\mathbb{X} \subseteq \mathbb{R}^d$ with a positive integer $d$ and the output space is $\mathbb{Y} \subseteq \mathbb{R}$. Let $\mathbb{D}^{\mathrm{tr}} = \{(\boldsymbol{x}_i^{\mathrm{tr}}, y_i^{\mathrm{tr}})\}_{i=1}^n$ be training samples drawn from a training distribution whose density is $p_{tr}(\boldsymbol{x}, y)$ in an i.i.d. fashion. A prediction model $f : \mathbb{X} \to \mathbb{Y}$ is trained using $\mathbb{D}^{\mathrm{tr}}$ in a training phase.

In an operational phase after the training phase, the prediction model $f$ is used to predict output values of corresponding inputs $\mathbb{U}^{\mathrm{op}} = \{\boldsymbol{x}^{\mathrm{op}}\}_{i=1}^m$ drawn from an operational distribution $p_{op}(\boldsymbol{x})$. The mean squared error (MSE) of $f$ over the joint operational distribution $p_{op}(\boldsymbol{x}, y)$ is defined as the expectation as follows:

$$\mathrm{MSE}(f) := \mathbb{E}_{p_{op}(\boldsymbol{x}, y)} \left[ (y - f(\boldsymbol{x}))^2 \right], \tag{1}$$

where $\mathbb{E}_{p(\boldsymbol{x}, y)}[g(\boldsymbol{x}, y)]$ computes the expectation of $g(\boldsymbol{x}, y)$ over the density $p(\boldsymbol{x}, y)$ as $\int_{\mathbb{X} \times \mathbb{Y}} g(\boldsymbol{x}, y) dp(\boldsymbol{x}, y)$. We intend to estimate $\mathrm{MSE}(f)$ after the prediction and before observing the actual values of target variables, i.e., estimate $\mathrm{MSE}(f)$ using $f, \mathbb{D}^{\mathrm{tr}}$, and $\mathbb{U}^{\mathrm{op}}$. However, MSE estimation without any assumption is infeasible. Hence, we employ the covariate shift assumption, which is a prevalent setting for machine learning in the wild. The problem definition is then formalized as follows.

**Definition 1** (MSE estimation problem). Given a regression model $f : \mathbb{X} \to \mathbb{Y}$ trained with training data $\mathbb{D}^{\mathrm{tr}} = \{(\boldsymbol{x}_i^{\mathrm{tr}}, y_i^{\mathrm{tr}})\}_{i=1}^n$, i.i.d. samples from a training distribution $p_{tr}(\boldsymbol{x}, y) = p_{tr}(\boldsymbol{x})p(y|\boldsymbol{x})$, and operational data $\mathbb{U}^{\mathrm{op}} = \{\boldsymbol{x}_i^{\mathrm{op}}\}_{i=1}^m$, i.i.d. samples from an operational distribution $p_{op}(\boldsymbol{x})$, the task is to estimate $\mathrm{MSE}(f)$ with $p_{op}(\boldsymbol{x}, y) = p_{op}(\boldsymbol{x})p(y|\boldsymbol{x})$.

### 2.2 RELATED WORKS

We categorize the related works into three groups: mean and variance estimation methods, uncertainty estimation methods, and accuracy estimation methods for classification.

**Mean and variance estimation methods.** Eq. (1) can be expanded as

$$\mathrm{MSE}(f) = \mathbb{E}_{p_{op}(\boldsymbol{x})} \left[ \mathbb{E}_{p(y|\boldsymbol{x})} \left[ (y - f(\boldsymbol{x}))^2 \right] \right] = \mathbb{E}_{p_{op}(\boldsymbol{x})} \left[ (u(\boldsymbol{x}) - f(\boldsymbol{x}))^2 + \sigma(\boldsymbol{x})^2 \right], \tag{2}$$

where we define $u(x) := \mathbb{E}_{p(y|\boldsymbol{x})}[y]$ and $\sigma(\boldsymbol{x})^2 := \mathbb{E}_{p(y|\boldsymbol{x})}[(y - u(\boldsymbol{x}))^2]$. Based on the expansion, one can estimate MSE by approximating both $u(\boldsymbol{x})$ and $\sigma(\boldsymbol{x})$. In other words, MSE can be estimated by a two-step approach; one step for computing $u(\boldsymbol{x})$ and the other step for computing $\sigma(\boldsymbol{x})$.

Nix & Weigend (1994) propose to train a mean and variance networks, where two networks each learns mean and variance, which jointly trained by maximizing the log likelihood for the training data. Skafte et al. (2019) further adopt (a) locally-aware mini-batching scheme with adjusted sample weights, (b) mean and variance split training, (c) estimating Inv-Gamma distribution for $\sigma(\boldsymbol{x})$ instead of the point estimation of $\sigma(\boldsymbol{x})$, and (d) extrapolation architecture. These methods successfully captures means and variances when abundant numbers of samples are available. However, as described in the introduction, the problem is that estimating means and variances is rather difficult problem than solely solving the MSE estimation problem. According to Vapnik's principle, this approach should be avoided.

**Uncertainty estimation methods.** Recently, uncertainty estimation methods have been actively studied in the field of machine learning (Abdar et al., 2021). Monte Carlo dropout (Gal & Ghahramani, 2016) employs dropout (Srivastava et al., 2014) as an approximation of Bayesian neural networks and is utilized for uncertainty estimation (Wang et al., 2019; Liu et al., 2019). Lakshminarayanan et al. (2017) proposed a technique called DeepEnsemble, in which neural networks are trained with varying random initializations and then ensembled to achieve a high capability for estimating uncertainty. Malinin et al. (2021) also investigated uncertainty estimation for gradient boosting models, proposing an ensemble method for gradient boosting decision trees. Gaussian processes (Rasmussen & Williams, 2005) are considered to be part of uncertainty estimation methods with built-in uncertainty estimation. These methods output the mean and uncertainty, which may be related to the variance. Consequently, we can use Eq. (2) to estimate the MSE. However, uncertainty in areas where $p(\boldsymbol{x})$ is small is often very high, capturing epistemic uncertainty (a.k.a knowledge uncertainty), and this high uncertainty may be unnecessary and potentially worsen the MSE estimation error.

**Accuracy estimation methods for classification.** Similarly to the MSE estimation problem, the accuracy estimation problem has been studied for classification tasks. Several works have proposed estimating accuracy based on the distribution shift of the input variables between the training and operational data (Elsahar & Gallé, 2019; Do et al., 2021; Deng & Zheng, 2021; Schelter et al., 2020; Techapanurak & Okatani, 2021). Chen et al. (2021b) employed domain adaptation methods for better estimation. Recently, the use of "check" models, which verify the predictions of the model, has been proposed and achieved superior performance in accuracy estimation for image and text classification tasks compared to the methods based on distribution shifts (Chuang et al., 2020; Chen et al., 2021a). These accuracy estimation methods cannot be directly applied to the MSE estimation problem due to the differences in the nature of classification and regression tasks, i.e., discrete versus continuous target variables.

## 3 PROPOSED METHOD

In this section, we introduce a new method to estimate MSE by directly minimizing the MSE estimation error. Furthermore, through our theoretical analysis, we present a generalization error bound for this approach.

### 3.1 UPPER BOUNDING THE MSE ESTIMATION ERROR

We estimate MSE by the expectation over $p(\boldsymbol{x})$ of the squared error between $h(\boldsymbol{x})$ and $f(\boldsymbol{x})$ as

$$\widehat{\mathrm{MSE}}(f; h) = \mathbb{E}_{p_{op}(\boldsymbol{x})} \left[ 2(h(\boldsymbol{x}) - f(\boldsymbol{x}))^2 \right], \tag{3}$$

which replace the variable $y$ in Eq. (1) with the output of another model $h$. We call $h$ as "check model" in this paper. The MSE estimation error $E(h)$ is then defined as the absolute error between the true MSE $\mathrm{MSE}(f)$ and the estimated MSE $\widehat{\mathrm{MSE}}(f; h)$.

$$E(h) := \left| \mathrm{MSE}(f) - \widehat{\mathrm{MSE}}(f) \right| \tag{4}$$

We aim at minimizing $E(h)$ with regard to $h$. In the followings, we derive a training objective for $h$ which directly minimizes $E(h)$. Our analysis is based on an inequality regarding the squared expectation and the expectation of the squared values as stated in Lemma 1.

**Lemma 1.** The following inequality holds:

$$\left(\mathbb{E}[x]\right)^2 \leq \mathbb{E}\left[s(x)x^2\right] \leq \mathbb{E}\left[x^2\right] \leq 2\,\mathbb{E}\left[s(x)x^2\right], \tag{5}$$

where $s(x) := \mathbf{1}_{(x \geq 0 \wedge \mathbb{E}[(x)^{+2}] \geq \mathbb{E}[(-x)^{+2}]) \vee (x < 0 \wedge \mathbb{E}[(x)^{+2}] < \mathbb{E}[(-x)^{+2}])}$ and $(x)^{+2} := \max(0, x)^2$ .

Note $\mathbf{1}_c$ denotes the indicator function and $\mathbf{1}_c = 1$ if the condition $c$ is true, otherwise 0. The proof is based on a direct calculation using Jensen's inequality and its details are presented in Appendix.

The MSE estimation error $E(h)$ can be rewritten as

$$E(h)^2 = \left|\mathbb{E}_{p_{op}(\boldsymbol{x},y)}\left[(y-f)^2\right] - \mathbb{E}_{p_{op}(\boldsymbol{x})}\left[2(h-f)^2\right]\right|^2 = \left(\mathbb{E}_{p_{op}(\boldsymbol{x},y)}\left[e_f(h,\boldsymbol{x},y)\right]\right)^2 \tag{6}$$

where $e_f(h, \boldsymbol{x}, y) := (y - f(\boldsymbol{x}))^2 - 2(h(\boldsymbol{x}) - f(\boldsymbol{x}))^2$ is the difference between the squared error of $f$ for sample $(\boldsymbol{x}, y)$ and its estimation $2(h(\boldsymbol{x}) - f(\boldsymbol{x}))^2$ computed by $h$. We apply Lemma 1 to Eq. (6) and obtain the following theorem.

**Theorem 2.** Let us define $K(h), K^+(h), K^-(h)$ and $K^*(h)$ as $K(h) = \mathbb{E}_{p_{op}(\boldsymbol{x},y)}\left[e_f(h,\boldsymbol{x},y)^2\right]$, $K^+(h) = \mathbb{E}_{p_{op}(\boldsymbol{x},y)}\left[(e_f(h,\boldsymbol{x},y))^{+2}\right]$, $K^-(h) = \mathbb{E}_{p_{op}(\boldsymbol{x},y)}\left[(-e_f(h,\boldsymbol{x},y))^{+2}\right]$, and $K^*(h) = \mathbb{E}_{p_{op}(\boldsymbol{x},y)}\left[s(e_f(h,\boldsymbol{x},y))e_f(h,\boldsymbol{x},y)^2\right]$, where $s(t) = \mathbf{1}_{(t \geq 0 \wedge K^+(h) \geq K^-(h)) \vee (t < 0 \wedge K^+(h) < K^-(h))}$. Then, the MSE estimation error $E(h)$ is bounded as

$$E(h)^2 \leq K^*(h) \leq K(h) \leq 2K^*(h). \tag{7}$$

We omit the proof since the theorem is obvious. Theorem 2 indicates that the MSE estimation error is upper bounded by $K^*(h)$ and this bound is tighter than the bound $K(h)$, which is obtained by the naive derivation by Jensen's inequality.

$K^*(h)$ is a promising objective function to train $h$ for the MSE estimation problem. In practice, we minimize the empirical version of $K^*(h)$. However, it requires samples $(\boldsymbol{x}^{\mathrm{op}}, y^{\mathrm{op}})$ over the operational density $p_{op}(\boldsymbol{x}, y)$ and the target variables $y^{\mathrm{op}}$ are unavailable by the definition of the problem. Hence, we exploit $\mathbb{D}^{\mathrm{tr}}$ to train $h$.[1] Note that using $\mathbb{D}^{\mathrm{tr}}$ instead for $\mathbb{D}^{\mathrm{op}}$ is practically valid under the absolutely continuous assumption where $p_{tr}(\boldsymbol{x}) = 0 \Rightarrow p_{op}(\boldsymbol{x}) = 0$ (Fang et al., 2020) and a model can globally fit to the data (Quionero-Candela et al., 2009). The training objective is thus defined as

$$\widehat{K}^*(h; \mathbb{D}^{\mathrm{tr}}) := \frac{1}{|\mathbb{D}^{\mathrm{tr}}|} \sum_{(\boldsymbol{x}, y \in \mathbb{D}^{\mathrm{tr}})} \widehat{s}(e_f(h, \boldsymbol{x}, y)) e_f(h, \boldsymbol{x}, y)^2, \tag{8}$$

where $\widehat{s}(t) := \mathbf{1}_{(t \geq 0 \wedge \widehat{K}^+(h;\mathbb{D}^{\mathrm{tr}}) \geq \widehat{K}^-(h;\mathbb{D}^{\mathrm{tr}})) \vee (x < 0 \wedge \widehat{K}^+(h;\mathbb{D}^{\mathrm{tr}}) < \widehat{K}^-(h;\mathbb{D}^{\mathrm{tr}}))}$ with

$$\widehat{K}^+(h; \mathbb{D}^{\mathrm{tr}}) := \frac{1}{|\mathbb{D}^{\mathrm{tr}}|} \sum_{(\boldsymbol{x}, y \in \mathbb{D}^{\mathrm{tr}})} (e_f(h, \boldsymbol{x}, y))^{+2}, \tag{9}$$

$$\widehat{K}^-(h; \mathbb{D}^{\mathrm{tr}}) := \frac{1}{|\mathbb{D}^{\mathrm{tr}}|} \sum_{(\boldsymbol{x}, y \in \mathbb{D}^{\mathrm{tr}})} (-e_f(h, \boldsymbol{x}, y))^{+2}. \tag{10}$$

We train $h$ by minimizing $\widehat{K}^*(h; \mathbb{D}^{\mathrm{tr}})$ and predict MSE using $\mathbb{U}^{\mathrm{op}}$ as

$$\widehat{\mathrm{MSE}}(f; h, \mathbb{U}^{\mathrm{op}}) := \frac{1}{|\mathbb{U}^{\mathrm{op}}|} \sum_{\boldsymbol{x}^{\mathrm{op}} \in \mathbb{U}^{\mathrm{op}}} 2\left(h(\boldsymbol{x}^{\mathrm{op}}) - f(\boldsymbol{x}^{\mathrm{op}})\right)^2. \tag{11}$$

**Remark 3.** Our method of training $h$ with minimizing $\widehat{K}^*(h; \mathbb{D}^{\mathrm{tr}})$ is consistent against any overfitting of $f$. Suppose that $f$ is fully overfitting to $\mathbb{D}^{\mathrm{tr}}$, i.e., for any $(\boldsymbol{x}, y) \in \mathbb{D}^{\mathrm{tr}}$, $f(\boldsymbol{x}) = y$ holds, and let $u(\boldsymbol{x})$ be $\mathbb{E}_{p(y|\boldsymbol{x})}[y]$ and $\sigma(x)$ be $\sqrt{\mathbb{E}_{p(y|\boldsymbol{x})}[(y - u(\boldsymbol{x}))^2]}$. On one hand, the MSE of $f$ is approximated as $\mathrm{MSE}(f) = \mathbb{E}_{p_{op}(\boldsymbol{x},y)}[(y - f(\boldsymbol{x}))^2] \approx \mathbb{E}_{p_{op}(\boldsymbol{x})p(y|\boldsymbol{x})p(y'|\boldsymbol{x})}[(y - y')^2] = 2\,\mathbb{E}_{p_{op}(\boldsymbol{x})}[\sigma(\boldsymbol{x})^2]$ since $f(\boldsymbol{x})$ can be considered to output a value $y'$ of $(\boldsymbol{x}', y')$ in $\mathbb{D}^{\mathrm{tr}}$ which approximately follows $p(y'|\boldsymbol{x})$.

---

[1] Covariate shift adaptation methods (Yamada et al., 2011; Kanamori et al., 2009; Zhang et al., 2020) can be used for better training of $h$ using $\mathbb{U}^{\mathrm{op}}$. However, it requires training $h$ for every time $\mathbb{U}^{\mathrm{op}}$ changes, which is costly for continuous monitoring in a typical MLOps situation.

On the other hand, $e_f(h, \boldsymbol{x}, y) = 2(h(\boldsymbol{x}) - y)^2$ and $\widehat{K}^*(h) = \frac{4}{|\mathbb{D}^{\mathrm{tr}}|} \sum_{(\boldsymbol{x}, y \in \mathbb{D}^{\mathrm{tr}})} (h(\boldsymbol{x}) - y)^4$. The minimization of $\widehat{K}^*(h)$ leads $h(\boldsymbol{x})$ to be close to $u(\boldsymbol{x})$ for each $\boldsymbol{x}$. Then the estimated MSE becomes $\widehat{\mathrm{MSE}}(f; h) = 2 \mathbb{E}_{p_{op}(\boldsymbol{x})}[(f(\boldsymbol{x}) - u(\boldsymbol{x}))^2] \approx 2 \mathbb{E}_{p_{op}(\boldsymbol{x})p(y'|\boldsymbol{x})}[(y' - u(\boldsymbol{x}))^2] = 2 \mathbb{E}_{p_{op}(\boldsymbol{x})}[\sigma(x)^2]$, which corresponds to the approximated MSE above. Hence, even when $f$ is overfitting, our method precisely estimates the MSE.

It should be noted that one of the most naive methods for one-step MSE estimation is to fit a check model $h'$ to the square errors of the model by minimizing $\frac{1}{n} \sum_{(\boldsymbol{x}, y) \in \mathbb{D}^{\mathrm{tr}}} ((y - f(\boldsymbol{x})^2 - h'(\boldsymbol{x}))^2$ and estimate MSE by $\mathbb{E}_{p_{op}(\boldsymbol{x})}[h'(\boldsymbol{x})]$. However, it is evident that $h'$ is biased. In an extreme case where $f$ is perfectly fitted to $\mathbb{D}^{\mathrm{tr}}$, $h'$ is trained to output 0 for any input, as $(y - f(\boldsymbol{x}))^2 = 0$ for any $(\boldsymbol{x}, y) \in \mathbb{D}^{\mathrm{tr}}$, and the estimated MSE is consistently 0. This estimation is incorrect unless there exists no new sample in $\mathbb{D}^{\mathrm{op}}$, i.e., $(\boldsymbol{x}, y) \in \mathbb{D}^{\mathrm{op}} \Rightarrow (\boldsymbol{x}, y) \in \mathbb{D}^{\mathrm{tr}}$. Thus, it is a clear advantage that our method does not assume anything regarding $f$ as Remark 3 states.

## 3.2 THEORETICAL ANALYSIS

In this subsection, we establish an upper bound of the generalization error of the proposed method using the Rademacher complexity (Koltchinskii, 2001). We assume $p_{tr}(\boldsymbol{x}) = p_{op}(\boldsymbol{x}) = p(\boldsymbol{x})$ for simplicity.

Firstly, we justify our method of minimizing $\widehat{K}^*(h; \mathbb{D}^{\mathrm{tr}})$ for the MSE estimation problem by Lemma 4.

**Lemma 4.** Let $f : \mathbb{X} \to \mathbb{Y}$ be a given regression model and $\mathcal{H}$ be a family of functions mapping from $\mathbb{X}$ to $\mathbb{Y}$. Assume that (a) there exists a constant $M > 0$ such that $|e_f(h, \boldsymbol{x}, y)| \leq M$ holds for every $h \in \mathcal{H}$ and $(\boldsymbol{x}, y) \in \mathbb{X} \times \mathbb{Y}$, (b) there exists some constant $H > 0$ such that $|h(\boldsymbol{x}) - f(\boldsymbol{x})| \leq H$ holds for every $h \in \mathcal{H}$ and $\boldsymbol{x} \in \mathbb{X}$, Then for any $\delta \in (0, 1)$, with probability $1 - \delta$ over the draw of an i.i.d. sample $S$ of size $n$ from $p(\boldsymbol{x}, y) = p(\boldsymbol{x})p(y|\boldsymbol{x})$, the following inequality holds for all $h \in \mathcal{H}$:

$$E(h)^2 \leq \widehat{K}^*(h; S) + 16HM \, \mathfrak{R}_n(\mathcal{H}) + M^2 \sqrt{\frac{\log \frac{1}{\delta}}{2n}}, \tag{12}$$

where $\mathfrak{R}_n(\mathcal{H})$ is the Rademacher complexity of $\mathcal{H}$ for the sampling of size $n$ from $p(\boldsymbol{x}, y)$.

The proof is based on Theorem 3.3 in (Mohri et al., 2018), the definition of the Rademacher complexity, and Ledoux-Talagrand contraction lemma (Ledoux & Talagrand, 2013), and presented in Appendix. Lemma 4 shows that minimizing $\widehat{K}^*(h; \mathbb{D}^{\mathrm{tr}})$ makes the upper bound of the MSE estimation error lower. Hence, our method solves the MSE estimation problem.

Next, we provide an upper bound of the generalization error of a check model $\widehat{h}$ which is obtained by our method. Before stating the theorem, we prepare two lemmas regarding $J(h)$.

**Lemma 5.** The following inequality holds for every $h$:

$$E(h)^2 \leq J(h) := \mathbb{E}_{p(\boldsymbol{x})} \left[ \left( \mathbb{E}_{p(y|\boldsymbol{x})}[(y - f(\boldsymbol{x}))^2] - 2(h(\boldsymbol{x}) - f(\boldsymbol{x}))^2 \right)^2 \right] \tag{13}$$

**Lemma 6.** The following equality holds for every $h$:

$$K(h) = J(h) + C_p, \tag{14}$$

where $C_p = \mathbb{E}_{p(\boldsymbol{x}, y)} \left[ (y - f(\boldsymbol{x}))^4 \right] - \mathbb{E}_{p(\boldsymbol{x})} \left[ \left( \mathbb{E}_{p(y|\boldsymbol{x})}[(y - f(\boldsymbol{x}))^2] \right)^2 \right] \geq 0$ does not depend on $h$.

**Theorem 7.** Suppose that the assumptions made in Lemma 4 holds. Let $h^*$ be a minimizer of $K(h)$ and $\widehat{h}$ be a minimizer of $\widehat{K}^*(h; S)$. Assume that (a) the minimizer of $\widehat{K}^*$ makes $\widehat{K}$ lower than the minimizer of $K$, i.e., $\widehat{K}(\widehat{h}; S) \leq \widehat{K}(h^*; S)$, and (b) the minimizer of $K$ makes $K^*$ lower than the minimizer of $\widehat{K}^*$, i.e., $K^*(h^*) \leq K^*(\widehat{h})$. Then for any $\delta \in (0, 1)$, with probability $1 - \delta$ over the draw of an i.i.d. sample $S$ of size $n$ from $p(\boldsymbol{x}, y)$, the following inequality holds:

$$E(\widehat{h})^2 \leq J(h^*) + 32HM \, \mathfrak{R}_n(\mathcal{H}) + 4M^2 \sqrt{\frac{\log \frac{4}{\delta}}{2n}}. \tag{15}$$

Suppose that we employ a model family $\mathcal{H}$ such that $\mathfrak{R}_n(\mathcal{H}) = \mathcal{O}(1/\sqrt{n})$. Then we have

$$E(\widehat{h}) \leq \sqrt{J(h^*)} + \mathcal{O}_p(1/\sqrt[4]{n}), \tag{16}$$

where $\mathcal{O}_p$ denotes the order in probability. This shows that the MSE estimation error of the proposed method decreases at a rate of $n^{-1/4}$. If the lowest value of $J$ among $\mathcal{H}$ is small, i.e., $J(h^*)$ is small (remind that the minimizer of $J$ and $K$ are identical by Lemma 6), this guarantees a good performance of the proposed method in theory.

### 3.3 PRACTICAL MODIFICATION: A REGULARIZATION FOR CONSISTENT FITTING

In practical cases, training $h$ with $\widehat{K}^*(h)$ may be suffered from unstable optimization due to the multi-modality of $e_f(h, \boldsymbol{x}, y)$; When $y \neq f(\boldsymbol{x})$, there exists two values of $h(\boldsymbol{x})$ that achieves $e_f(h, \boldsymbol{x}, y) = 0$ i.e., $e_f(h, \boldsymbol{x}, y) = 0 \Leftrightarrow h(\boldsymbol{x}) = f(\boldsymbol{x}) \pm \frac{1}{\sqrt{2}}|y - f(\boldsymbol{x})|$. Hence, the naive optimization of $\widehat{K}^*(h)$ possibly makes $h$ non-smooth, e.g., $h$ fits to the higher one of the two as $h(\boldsymbol{x}) = f(\boldsymbol{x}) + \frac{1}{\sqrt{2}}|y - f(\boldsymbol{x})|$ at $\boldsymbol{x}$, while at $\boldsymbol{x}'$, $h$ may fit to the lower one as $h(\boldsymbol{x}') = f(\boldsymbol{x}') - \frac{1}{\sqrt{2}}|y' - f(\boldsymbol{x}')|$. This inconsistency may worsen the MSE estimation error especially for the inputs between $\boldsymbol{x}$ and $\boldsymbol{x}'$.

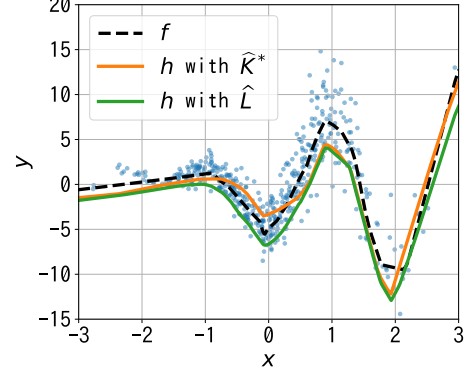

For better optimization of $h$, we encourage $h$ to always fit to the lower ones for every $\boldsymbol{x}$ by adding the following regularization term:

$$\widehat{R}(h; \mathbb{D}^{\mathrm{tr}}) := \frac{1}{|\mathbb{D}^{\mathrm{tr}}|} \sum_{(\boldsymbol{x}, y) \in \mathbb{D}^{\mathrm{tr}}} (f(\boldsymbol{x}) - y)^2$$
$$\times (h(\boldsymbol{x}) - (f(\boldsymbol{x}) - \varepsilon))^{+2}, \tag{17}$$

Figure 1: Effect of $\widehat{R}(h; \mathbb{D}^{\mathrm{tr}})$. We train $h$ with $\widehat{K}^*(h; \mathbb{D}^{\mathrm{tr}})$ and $\widehat{L}(h; \mathbb{D}^{\mathrm{tr}})$.[2] The regularized $h$ ($\widehat{L}(h; \mathbb{D}^{\mathrm{tr}})$) keeps lower than $f$, while the non-regularized one ($\widehat{K}^*(h; \mathbb{D}^{\mathrm{tr}})$) shows both higher (at $x = 0$) and lower (at $x = 1$) outputs than $f$.

where $\varepsilon \in \mathbb{R}$ is a small constant, e.g., $1.0 \times 10^{-3}$. $\widehat{R}(h; \mathbb{D}^{\mathrm{tr}})$ gives a penalty when $h(\boldsymbol{x})$ exceeds $(f(\boldsymbol{x}) - \varepsilon)$. Note $(f(\boldsymbol{x}) - y)^2$ disables the penalties when $y = f(\boldsymbol{x})$. The visualization of the effect of this regularization is demonstrated in Figure 1. Our regularization makes $h$ consistently lower than $h$ while $h$ trained solely with $\widehat{K}^*(h; \mathbb{D}^{\mathrm{tr}})$ fits to both higher and lower values than $f(x)$ among the inputs.

Our final objective function is,

$$\widehat{L}(h; \mathbb{D}^{\mathrm{tr}}) := \widehat{K}(h; \mathbb{D}^{\mathrm{tr}}) + \lambda \widehat{R}(h; \mathbb{D}^{\mathrm{tr}}), \tag{18}$$

where $\lambda \in \mathbb{R}_{\geq 0}$ is a hyperparameter whose default value can be 100. We use $\widehat{L}(h; \mathbb{D}^{\mathrm{tr}})$ to train $h$ and estimate the MSE using Eq. (11). The quantitative differences between the use of $\widehat{K}^*(h; \mathbb{D}^{\mathrm{tr}})$ and $\widehat{L}(h; \mathbb{D}^{\mathrm{tr}})$ are reported in the next section for both synthetic and real-world benchmark data sets, and this regularization is confirmed to improve the MSE estimation particularly for real-world data sets.

## 4 EXPERIMENTS

We conduct experiments on synthetic and benchmark data sets to verify the effectivity of the proposed method. The implementation is based on PyTorch (Paszke et al., 2019) and scikit-learn (Pedregosa et al., 2011). All experiments are carried out on a computational server equipping four Intel Xeon Platinum 8260 CPUs with 192 logical cores in total and 1TB RAM.

---

[2]The basic experimental setting follows subSection 4.1 while we use 500 training samples and 5-layer NN for $f$ and $h$ in order to clarify the effect of $\widehat{R}$.

### 4.1 EXPERIMENTS ON TOY DATA SETS

We first conduct experiments on synthetic toy data sets.

**Data.** We generate three synthetic data sets, A, B, and C, visualized in Figure 2. For all data sets, we generate the input variables $x \in \mathbb{R}$ from a normal distribution $\mathcal{N}(0, 1^2)$, where $\mathcal{N}(\mu, \sigma^2)$ denotes the Gaussian density with mean $\mu$ and variance $\sigma^2$. Then, we use the following equation to generate the corresponding target variable $y$ given $x$:

$$y = (3x + 5)\sin(3x + 5) + 0.3(1 + \max(0, 3x + 5))\varepsilon, \tag{19}$$

where $\varepsilon$ is a noise. For the synthetic A data, we generate the noise as a normal Gaussian random variable, i.e., $\varepsilon \sim \mathcal{N}(0, 1^2)$. For the synthetic B data, we employ the half-normal distribution, i.e., $\varepsilon = |\varepsilon'|$ where $\varepsilon' \sim \mathcal{N}(0, 1^2)$. Finally, for the synthetic C data, we use the Inv-Gamma distribution, i.e., $\varepsilon \sim \text{Inv-Gamma}(\alpha, \beta)$ where $\alpha$ is the shape parameter set to 2, and $\beta$ is the scale parameter set to 0.5. The synthetic B data is designed to assess the robustness of the MSE estimation methods against non-Gaussian noises, while the synthetic C data is used to evaluate the effect of outliers. Each data set consists of 100 training and 10,000 operational samples.

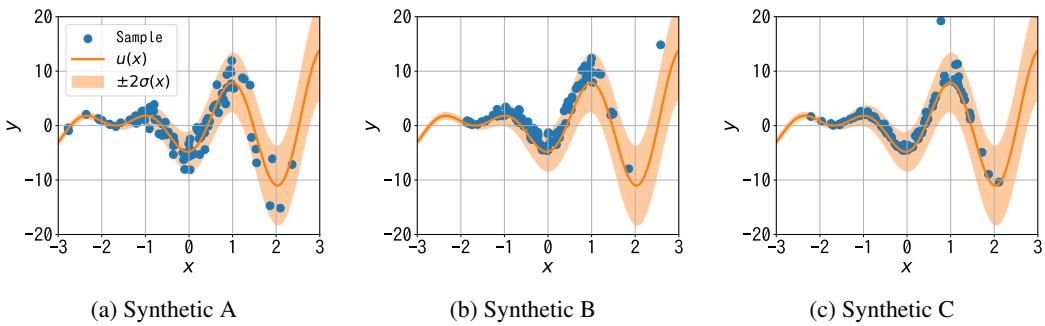

|  (a) Synthetic A | (b) Synthetic B | (c) Synthetic C |

Figure 2: Visualization of the synthetic data sets A, B, and C. The line of $u(x)$ denotes $y = (3x + 5)\sin(3x + 5)$ while $\sigma(x) = 0.3(1 + \max(0, 3x + 5))$ is the scale of noises.

**Setting.** We use a three-layer feedforward neural network (3-layer NN) for prediction models. The number of units in the hidden layer is set to 64, the activation function placed after the input and hidden layers is ReLU (Nair & Hinton, 2010). We train the network for 200 epochs with the Adam optimizer (Kingma & Ba, 2015) with its learning rate 0.01 and weight decay $1 \times 10^{-3}$ (Hanson & Pratt, 1988). The batch size is set to 100, i.e., we employ full batch training.

For estimating MSE, we train a 3-layer NN as $h$ to minimizes $\widehat{L}(h; \mathbb{D}^{\text{tr}})$ with 100.0. We also train $h$ with minimizing $\widehat{K}(h; \mathbb{D}^{\text{tr}})$ and $\widehat{K}^*(h; \mathbb{D}^{\text{tr}})$ to clarify the benefits of the use of $\widehat{K}^*$ and $\widehat{R}(h; \mathbb{D}^{\text{tr}})$. As a baseline, we use the following methods:

- [M2V] A naive baseline where a 3-layer NN is trained for mean by minimizing the training MSE first, and then another 3-layer NN is trained to estimate variance by maximizing the log likelihood, fixing the NN of mean estimation.
- [MVN] Mean and variance network (Nix & Weigend, 1994). We use two 3-layer NNs to estimate means and variances, which are jointly trained by maximizing the log likelihood.
- [ENS] DeepEnsemble (Lakshminarayanan et al., 2017). We train 10 MVN networks and compute the ensembled means and variances.
- [RVN] Reliable estimation of variance networks (Skafte et al., 2019). We train the mean estimation network for 100 epochs, and then train the mean and three variance-related networks for 10 epochs using their mini-batch strategy, which corresponds to 100 epochs of a standard training. The hyperparameters follow the proposed values in the paper. The implementation is based on codes provided by the author (Skafte, 2019).
- [RVNnE] A variant of RVN without its extrapolation architecture since epistemic uncertainty may worsen the results.
- [M2RVN] A variant of RVN where its mean network is fixed after the first 100 epochs.

The architecture and hyperparameters of NNs used in our method and baselines are the same with ones for prediction models described above.

Table 1: Average absolute MSE estimation errors (↓) for synthetic data sets over 100 trials. Numbers in brackets are the standard deviations. Boldface highlights the lowest error and comparable results based on the Wilcoxon signed-rank test (Wilcoxon, 1945) with a significance level of 5%. Asterisk and underline further spotlight the lowest and the second lowest errors, respectively.

| Data set | Baselines | | | | | | Ours | | |
|---|---|---|---|---|---|---|---|---|---|
| | M2V | MVN | ENS | RVN | RVNnE | M2RVN | $\widehat{K}$ | $\widehat{K}^*$ | $\widehat{L}$ |
| Synthetic A | 0.294 (0.259) | 1.411 (7.853) | 12.704 (116.765) | 1.010 (1.120) | 1.009 (1.121) | 0.553 (0.450) | 0.292 (0.231) | **0.235**$^*$ **(0.208)** | 0.259 (0.250) |
| Synthetic B | 0.307 (0.283) | 70.432 (676.069) | 325.730 (3143.811) | 0.940 (0.808) | 0.931 (0.798) | 0.560 (0.450) | 0.264 (0.201) | **0.252**$^*$ **(0.201)** | 0.259 (0.226) |
| Synthetic C | 0.751 (2.351) | 25.897 (190.993) | 118.872 (493.593) | 0.799 (0.724) | 0.819 (0.750) | 0.541 (0.411) | **0.407**$^*$ **(0.365)** | 0.412 (0.415) | **0.425** **(0.426)** |

As a data preprocess, the target variables $y$ in training and operational data are scaled based on the training data before being fed to the models. Then, MSE is computed and estimated for this scaled $y$. The evaluation is based on the absolute error of the empirical MSE and estimated MSE, where the empirical MSE is computed using the operational data $\mathbb{D}^{\mathrm{op}}$ as $\mathrm{MSE}(f, \mathbb{D}^{\mathrm{op}}) := \frac{1}{|\mathbb{D}^{\mathrm{op}}|} \sum_{(\boldsymbol{x}, y) \in \mathbb{D}^{\mathrm{op}}} (y - f(\boldsymbol{x}))^2$. We conduct experiments 100 times for each data set generated with different random seeds, and report the average errors. We also perform statistical tests to clarify the significance of the results.

**Result.** The results are presented in Table 1, displaying the average absolute MSE estimation errors over 100 trials. These findings demonstrate that none of the baselines achieve the best or even comparable results, highlighting the superiority of our one-step approach for MSE estimation. Our method minimizing $\widehat{K}^*$ achieves the lowest errors for synthetic A and B data, while the one minimizing $\widehat{K}$ is the most successful for synthetic C data. The results obtained with $\widehat{K}^*$ are either the second best or competitive. Although adding the regularization term $\widehat{R}$ does not improve MSE estimation compared to $\widehat{K}^*$, using $\widehat{L}$ still outperforms the naive upper bound $\widehat{K}$.

Regarding the baselines' results, MVN and ENS exhibit very high estimation errors for synthetic B and C data. Apart from our methods, M2V or M2RVN provide the lowest errors, indicating the effectiveness of separately training the mean and variance networks in a two-step process for these synthetic data sets.

In terms of robustness against different types of noise, MVN and ENS are weak against non-Gaussian noise (Synthetic B and C), M2V and our methods appear vulnerable to outliers (Synthetic C), while RVN and its variants are stable for all types of noise.

Further analyses, including qualitative evaluation and robustness of our method for overfitting and underfitting models are available in Appendix.

## 4.2 EXPERIMENTS ON BENCHMARK DATA SETS

We next perform experiments on benchmark data sets to demonstrate the usefulness of our method in real world.

**Data.** We use mg, space-ga, cpusmall, cadata, and abalone data sets obtained from LIBSVM data sets (Chang & Lin, 2023). The data set statistics are summarized in Table 2.

**Setting.** We select 1,000 continuous samples from a data set, starting with an index chosen uniformly at random. The first 500 samples are used as training data, while the remaining 500 samples serve as operational data. We train the prediction model, which is a 3-layer neural network (NN) using the Adam optimizer for 200 epochs. The number of hidden units, learning rate, weight

Table 2: Data set statistics.

| Name | #Samples | #Features |
|---|---|---|
| mg | 1385 | 6 |
| space-ga | 3107 | 6 |
| abalone | 4177 | 8 |
| cpusmall | 8192 | 12 |
| cadata | 20640 | 8 |

Table 3: Average absolute MSE estimation errors ($\downarrow$) for benchmark data sets over 100 trials. Numbers in brackets are the standard deviations. Boldface highlights the lowest error and comparable results based on the Wilcoxon signed-rank test with a significance level of 5%. Asterisk and underline further spotlight the lowest and the second lowest errors, respectively. NA indicates that error exceeds 1000.

| Data set | Baselines | | | | | | Ours | | |
|---|---|---|---|---|---|---|---|---|---|
| | M2V | MVN | ENS | RVN | RVNnE | M2RVN | $\widehat{K}$ | $\widehat{K}^*$ | $\widehat{L}$ |
| mg | 0.064 (0.030) | 0.050 (0.028) | 0.052 (0.027) | 0.130 (0.062) | 0.134 (0.053) | **0.044** (**0.034**) | 0.122 (0.022) | 0.072 (0.025) | **0.041**$^*$ (**0.026**) |
| space-ga | 0.389 (0.393) | 0.355 (0.380) | 0.345 (0.381) | 0.982 (0.333) | 0.968 (0.272) | 0.532 (0.213) | 0.314 (0.216) | 0.281 (0.202) | **0.278**$^*$ (**0.393**) |
| abalone | 0.995 (1.372) | 0.964 (1.379) | 0.962 (1.376) | **0.935** (**1.166**) | **0.971** (**1.201**) | 0.921 (**1.269**) | 0.992 (1.361) | 0.929 (1.314) | **0.919**$^*$ (**1.312**) |
| cpusmall | **21.329** (**212.062**) | NA | NA | 1.284 (0.258) | 1.291 (0.297) | 0.461 (0.094) | 0.045 (0.038) | 0.044 (0.038) | **0.028**$^*$ (**0.028**) |
| cadata | 56.048 (405.424) | NA | NA | **33.894** (**184.504**) | **24.318** (**105.585**) | 34.637 (119.868) | **9.086**$^*$ (**34.772**) | 11.560 (37.870) | **14.082** (**52.454**) |

decay, and batch size are tuned for each data set via a hyperparameter search by Optuna (Akiba et al., 2019). The hyperparameters can be found in Appendix.

The MSE estimation methods employed are the same as those used in the previous experiments, and the hyperparameters of the 3-layer NNs remain consistent with the ones tuned for each data set for the prediction models. We evaluate the absolute MSE estimation error between the empirical and estimated MSEs. The experiments are repeated 100 times with different starting indices and random seeds, and we report the average errors.

**Result.** The results are presented in Table 3. The table indicates that one of our proposed MSE estimation methods achieves the lowest error for each data set. This demonstrates the superiority of our one-step MSE estimation over conventional two-step estimation approaches for real-world data sets, validating the intuition of Vapnik's principle. M2V, MVN, and ENS are unstable for cpusmall and cadata, yielding very high MSE for some samples in the operational data, leading to high average and high standard deviations of errors. RVN and its variant RVNnE often perform worse than M2V, which is the most naive method. Comparing M2V (trains means then sigmas) vs MVN (jointly trains means and sigmas), and M2RVN (trains means then sigmas) vs RVN (iteratively trains means and sigmas), it appears that the joint and iterative training of mean and variance networks do not always outperform the two-step training.

Among the three variants of our proposed methods, firstly, the results achieved by $\widehat{K}^*$ are lower or comparable to those of $\widehat{K}$ since $\widehat{K}^*$ bounds $E^2$ more tightly and directly than $\widehat{K}$, as discussed in the previous section. Furthermore, the results obtained with $L$ are even better (or comparable) than those of $\widehat{K}^*$. This confirms that our regularization improves the learning of the check model $h$ for real-world data set.

## 5 CONCLUSION

In this work, we investigated the problem of MSE estimation. Instead of using existing two-step approaches, we proposed a novel one-step estimation method that directly and tightly upper bounds the MSE estimation error. Furthermore, we provided a theoretical upper bound for the generalization error of our method. Empirical results showed that our method yielded lower MSE estimation errors compared to the baseline methods for three synthetic and five benchmark data sets, thereby suggesting the superiority of our proposed method.

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
