APPENDIX (SUPPLEMENTARY MATERIAL)

## A  MISSING PROOFS

### A.1  PROOF OF LEMMA 1

**Proof.** Notice that $x = (x)^+ - (-x)^+$ and $x^2 = (x)^{+2} + (-x)^{+2}$, where $(x)^+ = \max(0, x)$.

$$|\mathbb{E}[x]|^2 = \left|\mathbb{E}\left[(x)^+ - (-x)^+\right]\right|^2 = \left|\mathbb{E}\left[(x)^+\right] - \mathbb{E}\left[(-x)^+\right]\right|^2 \tag{20}$$

$$\leq \max\left(\mathbb{E}\left[(x)^+\right], \mathbb{E}\left[(-x)^+\right]\right)^2 \tag{21}$$

$$\leq \max\left(\mathbb{E}\left[(x)^{+2}\right], \mathbb{E}\left[(-x)^{+2}\right]\right) \quad \text{(Jensen's inequality)} \tag{22}$$

$$\leq \mathbb{E}[(x)^{+2}] + \mathbb{E}[(-x)^{+2}] = \mathbb{E}\left[(x)^{+2} + (-x)^{+2}\right] = \mathbb{E}\left[x^2\right] \tag{23}$$

$$\leq 2\max\left(\mathbb{E}\left[(x)^{+2}\right], \mathbb{E}\left[(-x)^{+2}\right]\right) \tag{24}$$

If $\mathbb{E}[(x)^{+2}] \geq \mathbb{E}[(-x)^{+2}]$, we have

$$\mathbb{E}[s(x)x^2] = \mathbb{E}[\mathbf{1}_{x\geq 0}x^2] = \mathbb{E}[(x)^{+2}] = \max\left(\mathbb{E}[(x)^{+2}], \mathbb{E}[(-x)^{+2}]\right), \tag{25}$$

otherwise we have

$$\mathbb{E}[s(x)x^2] = \mathbb{E}[\mathbf{1}_{x<0}x^2] = \mathbb{E}[(-x)^{+2}] = \max\left(\mathbb{E}[(x)^{+2}], \mathbb{E}[(-x)^{+2}]\right). \tag{26}$$

Hence, $\mathbb{E}[s(x)x^2] = \max\left(\mathbb{E}[(x)^{+2}], \mathbb{E}[(-x)^{+2}]\right)$ holds, which concludes the proof combined with Eq. (24). ∎

### A.2  PROOF OF LEMMA 4

**Proof.** Based on Theorem 3.3 in (Mohri et al., 2018), for any $\delta \in (0, 1)$, with probability $1 - \delta$ over the draw of an i.i.d. sample $S$, the following holds:

$$K^*(h) \leq \frac{1}{n}\sum_{(\boldsymbol{x},y)\in S} s(e_f(h,\boldsymbol{x},y))e_f(h,\boldsymbol{x},y)^2 + 2\mathfrak{R}_n(\mathcal{S}) + M^2\sqrt{\frac{\log\frac{1}{\delta}}{2n}}, \tag{27}$$

$$\leq \widehat{K}^*(h; S) + 2\mathfrak{R}_n(\mathcal{S}) + M^2\sqrt{\frac{\log\frac{1}{\delta}}{2n}}, \tag{28}$$

where $\mathcal{S} = \{(\boldsymbol{x}, y) \mapsto s(e_f(h,\boldsymbol{x},y))e_f(h,\boldsymbol{x},y)^2 \mid h \in \mathcal{H}\}$ is the family of loss functions combined with $h \in \mathcal{H}$ and we use the fact that $\frac{1}{n}\sum_{(\boldsymbol{x},y)\in\mathbb{D}^{\text{tr}}} s(e_f(h,\boldsymbol{x},y))e_f(h,\boldsymbol{x},y)^2 \leq \frac{1}{n}\sum_{(\boldsymbol{x},y)\in S} \widehat{s}(e_f(h,\boldsymbol{x},y))e_f(h,\boldsymbol{x},y)^2 = \widehat{K}^*(h; S)$. Let $\phi : t \to s(t)t^2$, then $\mathcal{S} = \phi \circ \mathcal{E}$ where $\mathcal{E} = \{(\boldsymbol{x}, y) \mapsto e_f(h,\boldsymbol{x},y) \mid h \in \mathcal{H}\}$. Notice that $\phi$ is $2M$-Lipschitz for the input $e_f(h,\boldsymbol{x},y)$ for any $h, \boldsymbol{x}$, and $y$. By Ledoux-Talagrand contraction lemma (Ledoux & Talagrand, 2013), we have.

$$\mathfrak{R}_n(\mathcal{S}) = \mathfrak{R}_n(\phi \circ \mathcal{E}) \leq 2M\,\mathfrak{R}_n(\mathcal{E}) \tag{29}$$

$$\mathfrak{R}_n(\mathcal{E}) = \mathbb{E}_S\left[\mathbb{E}_{\sigma_i\in\{-1,1\}}\left[\sup_{h\in\mathcal{H}}\frac{1}{n}\sum_{i=1}^n \sigma_i e_f(h,\boldsymbol{x}_i, y_i)\right]\right] \tag{30}$$

$$= 2\,\mathbb{E}_S\left[\mathbb{E}_{\sigma_i\in\{-1,1\}}\left[\sup_{h\in\mathcal{H}}\frac{1}{n}\sum_{i=1}^n \sigma_i(h(\boldsymbol{x}) - f(\boldsymbol{x}))^2\right]\right] \tag{31}$$

$$\leq 4H\,\mathfrak{R}_n(\mathcal{H}), \tag{32}$$

where $\{\sigma_i\}_{i=1}^n$ is the Rademacher variables. Combined all of these, we conclude the proof. ∎

## A.3 Proof of Lemma 5

**Proof.** The proof is based on a direct calculation using Jensen's inequality.

$$E(h)^2 = \left| \mathbb{E}_{p(\boldsymbol{x},y)} \left[ (y - f)^2 \right] - \mathbb{E}_{p(\boldsymbol{x})} \left[ 2(h(\boldsymbol{x}) - f(\boldsymbol{x}))^2 \right] \right|^2 \tag{33}$$

$$= \left| \mathbb{E}_{p(\boldsymbol{x})} \left[ \mathbb{E}_{p(y|\boldsymbol{x})} \left[ (y - f(\boldsymbol{x}))^2 \right] - 2(h(\boldsymbol{x}) - f(\boldsymbol{x}))^2 \right] \right|^2 \tag{34}$$

$$\leq \mathbb{E}_{p(\boldsymbol{x})} \left[ \left( \mathbb{E}_{p(y|\boldsymbol{x})} \left[ (y - f(\boldsymbol{x}))^2 \right] - 2(h(\boldsymbol{x}) - f(\boldsymbol{x}))^2 \right)^2 \right] = J(h) \tag{35}$$

∎

## A.4 Proof of Lemma 6

**Proof.** Let $k(h)$ and $j(h)$ be;

$$k(h) := \mathbb{E}_{p(y|\boldsymbol{x})} \left[ \left( (y - f(\boldsymbol{x}))^2 - 2(h(\boldsymbol{x}) - f(\boldsymbol{x}))^2 \right)^2 \right] \tag{36}$$

$$j(h) := \left( \mathbb{E}_{p(y|\boldsymbol{x})} [(y - f(\boldsymbol{x}))^2] - 2(h(\boldsymbol{x}) - f(\boldsymbol{x}))^2 \right)^2 \tag{37}$$

Then, we have

$$k(h) = \mathbb{E}_{p(y|\boldsymbol{x})} \left[ \left( (y - f(\boldsymbol{x}))^2 - 2(h(\boldsymbol{x}) - f(\boldsymbol{x}))^2 \right)^2 \right] \tag{38}$$

$$= \mathbb{E}_{p(y|\boldsymbol{x})} \left[ (y - f(\boldsymbol{x}))^4 \right] - 4(h(\boldsymbol{x}) - f(\boldsymbol{x}))^2 \, \mathbb{E}_{p(y|\boldsymbol{x})} \left[ (y - f(\boldsymbol{x}))^2 \right] + 4(h(\boldsymbol{x}) - f(\boldsymbol{x}))^4 \tag{39}$$

$$j(h) = \left( \mathbb{E}_{p(y|\boldsymbol{x})} [(y - f(\boldsymbol{x}))^2] - 2(h(\boldsymbol{x}) - f(\boldsymbol{x}))^2 \right)^2 \tag{40}$$

$$= \left( \mathbb{E}_{p(y|\boldsymbol{x})} [(y - f(\boldsymbol{x}))^2] \right)^2 - 4(h(\boldsymbol{x}) - f(\boldsymbol{x}))^2 \, \mathbb{E}_{p(y|\boldsymbol{x})} [(y - f(\boldsymbol{x}))^2] + 4(h(\boldsymbol{x}) - f(\boldsymbol{x}))^4 \tag{41}$$

$$\therefore k(h) = j(h) + \mathbb{E}_{p(y|\boldsymbol{x})} \left[ (y - f(\boldsymbol{x}))^4 \right] - \left( \mathbb{E}_{p(y|\boldsymbol{x})} [(y - f(\boldsymbol{x}))^2] \right)^2 \tag{42}$$

Note by Jensen's inequality, $\mathbb{E}_{p(y|\boldsymbol{x})} \left[ (y - f(\boldsymbol{x}))^4 \right] - \left( \mathbb{E}_{p(y|\boldsymbol{x})} [(y - f(\boldsymbol{x}))^2] \right)^2 \geq 0$. By taking the expectation of $k(h)$ and $j(h)$ over $p(\boldsymbol{x})$, we have the following as desired.

$$K(h) = \mathbb{E}_{p(\boldsymbol{x})}[k(h)] = \mathbb{E}_{p(\boldsymbol{x})} \left[ j(h) + \mathbb{E}_{p(y|\boldsymbol{x})} \left[ (y - f(\boldsymbol{x}))^4 \right] - \left( \mathbb{E}_{p(y|\boldsymbol{x})} [(y - f(\boldsymbol{x}))^2] \right)^2 \right] \tag{43}$$

$$= \mathbb{E}_{p(\boldsymbol{x})}[j(h)] + \mathbb{E}_{p(\boldsymbol{x},y)} \left[ (y - f(\boldsymbol{x}))^4 \right] - \mathbb{E}_{p(\boldsymbol{x})} \left[ \left( \mathbb{E}_{p(y|\boldsymbol{x})} [(y - f(\boldsymbol{x}))^2] \right)^2 \right] \tag{44}$$

$$= J(h) + C_p \tag{45}$$

∎

## A.5 Proof of Theorem 7

Before the proof of Theorem 7, we establish a lemma below.

**Lemma A.** Suppose that the assumptions made in Lemma 4 holds. Then for any $\delta \in (0, 1)$, with probability $1 - \delta$ over the draw of an i.i.d. sample $S$ of size $n$ from $p(\boldsymbol{x}, y) = p(\boldsymbol{x})p(y|\boldsymbol{x})$, the following inequality holds for all $h \in \mathcal{H}$:

$$K(h) \leq \widehat{K}(h; S) + 16HM \, \mathfrak{R}_n(\mathcal{H}) + M^2 \sqrt{\frac{\log \frac{1}{\delta}}{2n}}, \tag{46}$$

where $\mathfrak{R}_n(\mathcal{H})$ is the Rademacher complexity of $\mathcal{H}$ for the sampling of size $n$ from $p(\boldsymbol{x}, y)$.

**Proof.** Based on Theorem 3.3 in (Mohri et al., 2018), for any $\delta \in (0, 1)$, with probability $1 - \delta$ over the draw of an i.i.d. sample $S$, the following holds:

$$\mathbb{E}_{p_{op}(\boldsymbol{x},y)} \left[ e_f(h, \boldsymbol{x}, y)^2 \right] \leq \frac{1}{n} \sum_{(\boldsymbol{x},y) \in \mathbb{D}^{tr}} e_f(h, \boldsymbol{x}, y)^2 + 2\mathfrak{R}_n(\mathcal{L}) + M^2 \sqrt{\frac{\log \frac{1}{\delta}}{2n}}, \tag{47}$$

$$\text{i.e.,} \quad K(h) \leq \widehat{K}(h; \mathbb{D}^{tr}) + 2\mathfrak{R}_n(\mathcal{L}) + M^2 \sqrt{\frac{\log \frac{1}{\delta}}{2n}}, \tag{48}$$

where $\mathcal{L} = \{(\boldsymbol{x}, y) \mapsto e_f(h, \boldsymbol{x}, y)^2 \mid h \in \mathcal{H}\}$ is the family of loss functions combined with $h \in \mathcal{H}$ and $\widehat{K}(h; S) := \frac{1}{n} \sum_{(\boldsymbol{x},y) \in S} e_f(h, \boldsymbol{x}, y)^2$ is the empirical version of $K(h)$. By Ledoux-Talagrand contraction lemma (Ledoux & Talagrand, 2013), we have

$$\mathfrak{R}_n(\mathcal{L}) = \mathbb{E}_S \left[ \mathbb{E}_{\sigma_i \in \{-1,1\}} \left[ \sup_{h \in \mathcal{H}} \frac{1}{n} \sum_{i=1}^n \sigma_i e_f(h, \boldsymbol{x}_i, y_i)^2 \right] \right] \tag{49}$$

$$\leq 2M \, \mathbb{E}_S \left[ \mathbb{E}_{\sigma_i \in \{-1,1\}} \left[ \sup_{h \in \mathcal{H}} \frac{1}{n} \sum_{i=1}^n \sigma_i e_f(h, \boldsymbol{x}_i, y_i) \right] \right] \tag{50}$$

$$= 4M \, \mathbb{E}_S \left[ \mathbb{E}_{\sigma_i \in \{-1,1\}} \left[ \sup_{h \in \mathcal{H}} \frac{1}{n} \sum_{i=1}^n \sigma_i (h(\boldsymbol{x}) - f(\boldsymbol{x}))^2 \right] \right] \tag{51}$$

$$\leq 8HM \, \mathfrak{R}_n(\mathcal{H}), \tag{52}$$

where $\{\sigma_i\}_{i=1}^n$ is the Rademacher variables. We conclude the proof by combining Eq. (48) with Eq. (52). ∎

We provide the proof of Theorem 7 using the lemma above.

**Proof of Theorem 7.** Based on Lemma 5 and Lemma 6, we have

$$E(\widehat{h})^2 - J(h^*) \leq J(\widehat{h}) - J(h^*) = K(\widehat{h}) - K(h^*)$$

$$= K(\widehat{h}) + \left( \widehat{K}(h^*; S) - \widehat{K}(h^*; S) \right) + \left( \widehat{K}^*(h^*; S) - \widehat{K}^*(h^*; S) \right) + \left( K^*(\widehat{h}) - K^*(\widehat{h}) \right)$$

$$+ \left( \widehat{K}^*(\widehat{h}; S) - \widehat{K}^*(\widehat{h}; S) \right) - K(h^*) \tag{53}$$

$$= \left( K(\widehat{h}) - \widehat{K}(h^*; S) \right) + \left( \widehat{K}^*(h^*; S) - K^*(\widehat{h}) \right) + \left( K^*(\widehat{h}) - \widehat{K}^*(\widehat{h}; S) \right)$$

$$+ \left( \widehat{K}^*(\widehat{h}; S) - \widehat{K}^*(h^*; S) \right) + \left( \widehat{K}(h^*; S) - K(h^*) \right) \tag{54}$$

$$= \text{(I)} + \text{(II)} + \text{(III)} + \text{(IV)} + \text{(V)}, \tag{55}$$

where;

$$\text{(I)} = K(\widehat{h}) - \widehat{K}(h^*; S) \tag{56}$$

$$\text{(II)} = \widehat{K}^*(h^*; S) - K^*(\widehat{h}) \tag{57}$$

$$\text{(III)} = K^*(\widehat{h}) - \widehat{K}^*(\widehat{h}; S) \tag{58}$$

$$\text{(IV)} = \widehat{K}^*(\widehat{h}; S) - \widehat{K}^*(h^*; S) \tag{59}$$

$$\text{(V)} = \widehat{K}(h^*; S) - K(h^*) \tag{60}$$

We bound the above five terms as follows: Based on the assumption and Lemma A, the following bounds hold with probability $1 - \delta/4$,

$$\text{(I)} = K(\widehat{h}) - \widehat{K}(h^*; S) \leq K(\widehat{h}) - \widehat{K}(\widehat{h}; S) \leq 16HM \, \mathfrak{R}_n(\mathcal{H}) + M^2 \sqrt{\frac{\log \frac{4}{\delta}}{2n}} \tag{61}$$

By the assumption and Hoeffding's inequality, we have the following inequality with probability $1 - \delta/4$:

$$\text{(II)} = \widehat{K}^*(h^*; S) - K^*(\widehat{h}) \leq \widehat{K}^*(h^*; S) - K^*(h^*) \leq M^2 \sqrt{\frac{\log \frac{4}{\delta}}{2n}} \tag{62}$$

Based on Lemma 4, with probability $1 - \delta/4$, we have

$$\text{(III)} = K^*(\widehat{h}) - \widehat{K}^*(\widehat{h}; S) \leq 16HM \, \mathfrak{R}_n(\mathcal{H}) + M^2 \sqrt{\frac{\log \frac{4}{\delta}}{2n}} \tag{63}$$

By the definition of $\widehat{h}$,

$$\text{(IV)} = \widehat{K}^*(\widehat{h}; S) - \widehat{K}^*(h^*; S) \leq 0 \tag{64}$$

Hoeffindg's inequality indicates that the following holds with probability $1 - \delta/4$;

$$\text{(V)} = \widehat{K}(h^*; S) - K(h^*) \leq M^2 \sqrt{\frac{\log \frac{4}{\delta}}{2n}} \tag{65}$$

Finally, we use the union bound to combine five inequalities from Eq. (61) to (65) to yields with probability $1 - \delta$

$$E(\widehat{h})^2 - J(h^*) \leq 32 HM \mathfrak{R}_n(\mathcal{H}) + 4M^2 \sqrt{\frac{\log \frac{4}{\delta}}{2n}}. \tag{66}$$

■

# B    EXPERIMENTAL DETAILS

## B.1    HYPERPARAMETERS USED IN SECTION 4.2

The hyperparameters of 3-layer NN used in Section 4.2 are summarized in Table B.1. These hyperparameters are tuned so that the prediction model with these hyperparameters achieves the lowest MSE on average.

Table B.1: Hyperparameters for 3-layer NNs.

| Data set | Hidden units | Learning rate | Weight decay | Batch size |
|---|---|---|---|---|
| mg | 128 | 1e-03 | 1e-08 | 50 |
| space-ga | 32 | 5e-03 | 1e-02 | 100 |
| cpusmall | 32 | 1e-02 | 1e-02 | 100 |
| cadata | 128 | 5e-04 | 1e-07 | 250 |
| abalone | 64 | 5e-04 | 1e-05 | 50 |

## B.2    MSE ESTIMATION FOR OVERFITTING MODELS.

In Remark 3, we claim that our method can estimate MSE even the model $f$ is overfitted to the training data. We empirically examine how our method estimate MSE for over- and under-fitting models.

**Setting.** We utilize the synthetic A data from Section 4.1 and increase the number of training samples to 500 to clearly discern the differences between overfitting and underfitting. For our prediction model, we employ a decision tree (DT) and vary the min-samples-split parameter from 1 to 100. It is important to note that, as illustrated in Figure B.1, setting the min-samples-split value to 1 causes the prediction model to remember all of the training data, leading to overfitting, while setting it to 100 results in underfitting. We train our method with $\widehat{K}$, $\widehat{K}^*$, and $\widehat{L}$ to estimate MSE. The experiment is repeated 30 times and we report the average errors.

**Result.** Figure B.2 illustrates the relationship between MSE and the MSE estimation errors for various min-samples-split settings. As shown in Figure B.2(a), our method can estimate the accuracy for any given setting, and the differences between the true MSE and the estimated MSE remain relatively stable on average, particularly in the case of $\widehat{K}^*$ and $\widehat{L}$. Figure B.2(b) suggests that the MSE estimation error may increase during overfitting and underfitting. However, prediction models are typically trained to avoid overfitting or underfitting the training data. Therefore, in practice, our methods are expected to achieve good MSE estimation errors.

We should once again note that a naive approach to fit $h'$ to $(y - f(\boldsymbol{x}))^2$ for $(\boldsymbol{x}, y) \in \mathbb{D}^{\text{tr}}$ would result in an estimated MSE of 0, which is most likely incorrect. Therefore, our method's ability to estimate MSE even for an overfitted model, as empirically demonstrated in Figure B.2, is remarkably impressive.

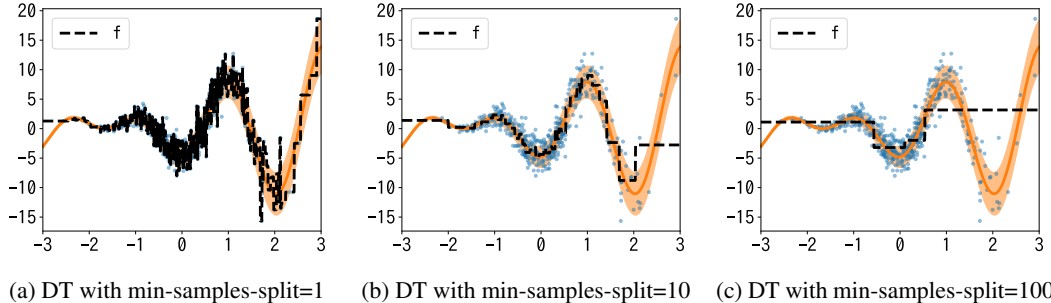

(a) DT with min-samples-split=1    (b) DT with min-samples-split=10    (c) DT with min-samples-split=100

Figure B.1: Decision trees (DTs) with different settings of min-samples-split. (a) shows an overfitting model while (c) shows an underfitting model. (b) is the a good fitting with low MSE over the operational data.

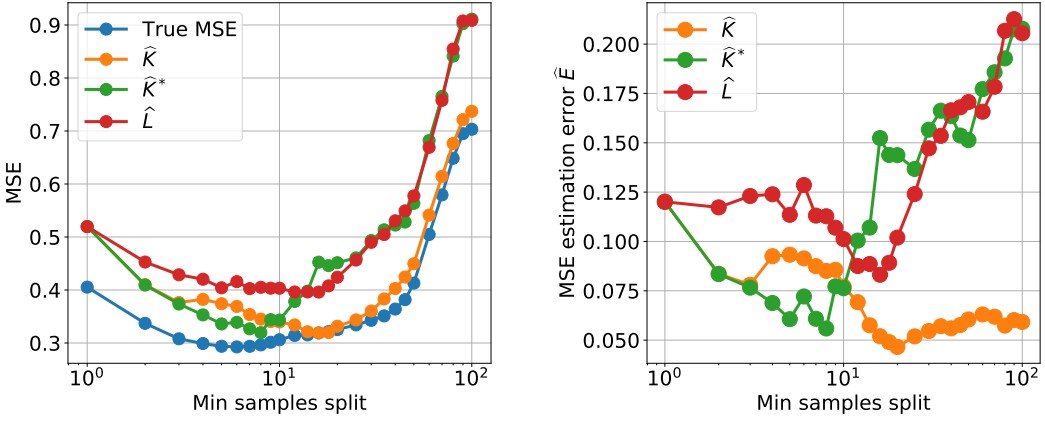

(a) Average true and estimated MSE against min-samples-split.

(b) Average MSE estimated errors against of min-samples-split.

Figure B.2: MSE estimation results with different settings of min-samples-split.

### B.3 QUALITATIVE EVALUATION.

Figure B.3 visualizes the models fitted by the MSE estimation methods, including the baselines and our methods, for experiments on a synthetic data set A, as described in Section 4.1. The fitted model by M2V, MVN, and ENS are all similar, with M2V and ENS displaying higher standard deviations than MVN at $x \leq -1$. Moreover, they exhibit very high standard variance for $x > 1$, compared to the true standard deviations. This leads to unnecessarily high MSE errors, resulting in significantly worse MSE estimation errors.

RVN and its variants fail to fit the data adequately. Since these methods train three to four 3-layer NNs which affect each other, 100 training data samples might not be sufficient for effective training.

For this data set, the difference between our methods employing $\widehat{K}$ and $\widehat{K}^*$ is limited. However, the effect of our regularization $\widehat{R}$ is evident in the result of $\widehat{L}$. The trained $h$ outputs lower values than those produced using $\widehat{K}$ and $\widehat{K}^*$, which makes its MSE estimation error the lowest among all of these methods.

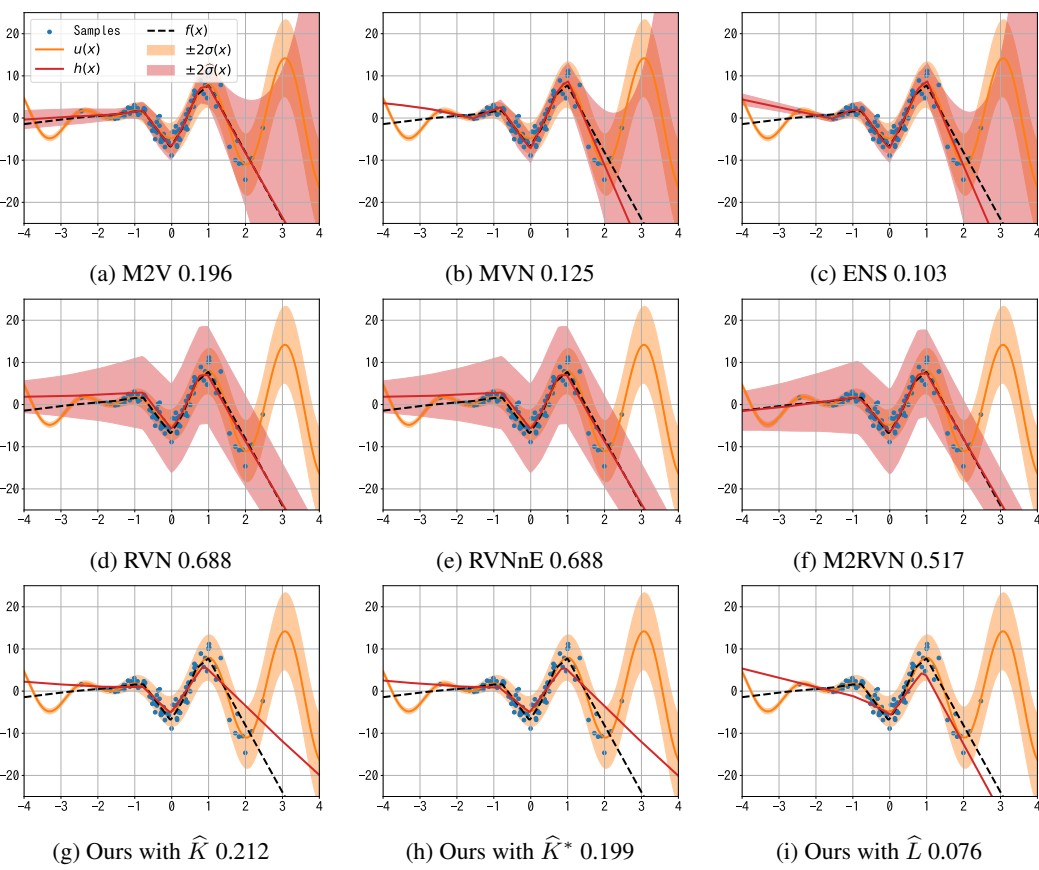

Figure B.3: Visualization of fitted models by MSE estimation methods for the synthetic A data set. The numbers in the captions are the MSE estimation errors. For baselines, we visualize the estimated standard deviations.