# OpenReview forum: "A One-Step MSE Estimation of Models in Production"
_ICLR.cc/2024/Conference — ICLR 2024 Conference Withdrawn Submission_

### Official Review · Reviewer_ZbmE · 2023-10-25

**Soundness:** 2 fair
**Presentation:** 3 good
**Contribution:** 2 fair
**Rating:** 5
**Confidence:** 4

**Summary:**

Model prediction evaluation metrics are important in evaluating the accuracy of the model. In particular, using models to evaluate other models is still a nascent field, and in particular have thus far been not well explored in context of regression as compared to classification. The authors suggest using a one-step method of estimating the MSE compared to other methods which evaluate both the mean and the variance.

**Strengths:**

The authors prove good bounds as compared to a naive Jensen's approach for one-step MSE estimation. Then the authors demonstrate superior MSE estimation powers for a synthetic dataset and some real-world datasets as well.

**Weaknesses:**

Unfortunately, the dataset choice is non-standard. Other baselines that the authors conduct their experiments against all choose to use the standard UCI datasets which are more commonly integrated, whereas the authors chose to use LIBSVM here. Some experiments on the more standard datasets would be appreciated.

Use of "tightly" would suggest that the upper bound proposed is tight, but it seems like the authors just prove that it is tighter than that of a naive bound based off of Jensen's.

Other
Minor typos:
overffiting

**Questions:**

Why 1000 samples for the datasets in particular? Seems like authors are trying to tackle the case for when it is not true that "abundant numbers of samples are available", but that doesn't seem to be part of the main points.
If that's the case, then also checking results on the full dataset would also be helpful to prove the point, especially since the paper claims that their objective is consistent against overfitting.

---

### Official Review · Reviewer_TPbn · 2023-10-30

**Soundness:** 2 fair
**Presentation:** 2 fair
**Contribution:** 2 fair
**Rating:** 3
**Confidence:** 2

**Summary:**

The paper introduces a new MSE estimation method. Instead of learning the mean and variance of the outputs, the proposed method learns a check model which gives pseudo labels for the MSE estimation. Generalization bounds and experiments have been provided to justify the method.

**Strengths:**

-	The paper provides both theoretical and empirical demonstration for the proposed method.
-	The empirical improvements in real-world data seem to be significant.

**Weaknesses:**

- The clarity needs improvement.
- The comparison to existing work needs to be clearer.

See "Questions" for details.

**Questions:**

-	The argument of Vapnik’s principle is vague. Why is mean and variance estimation more complicated than MSE? Why is the proposed method less complicated than previous methods? The argument needs to be more rigorous.
-	The authors claimed that the disadvantage of uncertainty estimation methods is the high uncertainty for small data regions. “Uncertainty in areas where p(x) is small is often very high, capturing epistemic uncertainty (a.k.a knowledge uncertainty), and this high uncertainty may be unnecessary and potentially worsen the MSE estimation error.” I do not follow the argument here. Does it mean we should not consider epistemic uncertainty?
-	What data does Eq.4 compute over? Does the training of the check model require a validation set? If so, does the check model just try to fit the validation set?
-	Why do existing estimation methods for classification not apply to regression? The paper only mentions they are not applicable since classification has discrete outputs and regression has continuous outputs. It is still unclear why they are not applicable.
-	It is unclear what the differences are between the proposed check model method and existing check model methods.
-	What if the variance of the small data region (x in [2,3]) is indeed very large. Will the proposed method still be better than uncertainty estimation methods?
-	Is the proposed method affected by the performance of f(x)? In other words, will the MSE estimation be accurate for both good and bad models f?

---

### Official Review · Reviewer_q3T4 · 2023-11-04

**Soundness:** 2 fair
**Presentation:** 3 good
**Contribution:** 2 fair
**Rating:** 3
**Confidence:** 3

**Summary:**

This work proposes a novel estimator for the MSE of a predictor, $E(f(x)-y)^2$, in a setting where we do not have access to test outputs and the input may exhibit covariate shift.  The method trains a "check model" $h(x)$ on training data to minimize the error $((E(f(x)-y)^2-E(f(x)-h(x))^2)^2$. The authors argued that the estimator has a non-degenerate behavior when $f$ is overfitted on training data, and the generalization error of $h$ can be controlled when its function class has a controlled Rademacher complexity.  Empirically, the proposed method outperform various instantiations of a naive estimator based on conditional mean and variance estimations.

**Strengths:**

- The paper is generally well-written.
- The proposed method demonstrates promising performance in experiments.

**Weaknesses:**

I am concerned about the validity and relevance of various theoretical claims which are used to motivate the work.

**1.** Section 3.1 claims that the proposed estimator is more robust against the overfitting of $f$ because when $f(x_i)\equiv y_i$ for all training samples $(x_i,y_i)$, the proposed estimator has a more sensible behavior, whereas a standard MSE estimator based on estimating $E(f(x)-y\mid x)$ will output 0.  However,
- it is evident that any reasonable implementation of the latter should estimate $E(f(x)-y\mid x)$ on a held-out set, in which case the pathology will not appear.
- And the behavior of the proposed estimator is not necessarily more sensible without a similar sample splitting: as the authors note, the proposed objective reduces to $E((h(x)-y)^4)$.  This is not necessarily well-defined, for example when $y$ does not have a bounded 4th moment.  And even when it is, there is no guarantee that the minimizer is $h^*(x)=E(y|x)$, as the authors claimed (consider any skewed error distribution).

**2.** Section 3.2 shows that the MSE estimate may enjoy a controlled generalization error if the hypothesis space for $h$ has a controlled Rademacher complexity.  It is unclear why such a hypothesis space cannot be adapted to estimate the mean and variance for the traditional ("two-step") MSE estimator.  It is true that strictly speaking, the estimation targets are different functions, so the approximation errors can be different.  But it appears to me that it is more reasonable to assume the approximation error for mean and variance estimation is at least not larger than that for $h$, as the former appears to be more natural targets.

**Questions:**

Clarifications to point (1) above would be welcome.

For (2), I think a comparison of convergence rates for different estimators would be necessary if the authors want to claim theoretical benefits.

---

### Official Review · Reviewer_YTwt · 2023-11-04

**Soundness:** 2 fair
**Presentation:** 2 fair
**Contribution:** 2 fair
**Rating:** 3
**Confidence:** 3

**Summary:**

The authors propose a method for estimating the MSE under the covariate shift assumption. The method relies on replacing the labels with a check model that is trained to minimize the mismatch with the true MSE.

**Strengths:**

- The writing flows reasonable well in several parts of the paper and the technical sections are reasonably clear
- The method is simple and easy to implement
- The method comes with rigorous theoretical analysis

**Weaknesses:**

- First, I think some aspects of the presentation (especially the abstract and intro) can still be greatly improved. It took me a while to understand what problem the authors were trying to solve. The key statement of "estimating MSE under covariate shift" is buried in the preliminaries section inside a long paragraph. The words "covariate shift" are not found anywhere in the abstract or intro as far as I can tell.
- Second of all, the experimental results are on the weak side. The datasets being used are simple (UCI datasets), which is potentially okay, but the results are not very strong. Out of five datasets, on at least two there seems to essentially no benefit from the method, and on two others there are for some reason massive errors bars around the metric, which makes it hard to understand what's going and if there is a true improvement.
- Most importantly, I am confused by why this approach (and the baselines) are being used instead of a simpler approaches based on importance sampling. When seeing the problem, my first reaction is that if I have data (x,y) from distribution p_1 and unlabeled data x from distribution p_2, and I want to estimate error under p_2, than I can compute an importance sampled estimate of the MSE on data from p_1 using p_2/p_1 as my importance weights (these ratios of density can be estimated via supervised learning). I am confused by why this simple approach is not being discussed at all and why these other approaches (which to me appear less standard) are used.
- A follow-up on the above, is that methods for domain adaptation like importance sampling (and there are many others) are not part of the baselines.

**Questions:**

- Why are some error bars so large and can there be more experimental evidence that the method works?
- Why would one use this method instead of importance sampling?
- Can importance sampling be a baseline?

---

### Official Review · Reviewer_Mmfr · 2023-11-05

**Soundness:** 2 fair
**Presentation:** 2 fair
**Contribution:** 2 fair
**Rating:** 3
**Confidence:** 2

**Summary:**

This paper studies how to measure and optimize models
trained for regression on p_tr for a downstream MSE
on heldout samples from an operational distribution p_op.
This setup is summarized in def 1.
The related work is on models that estimate the
mean and variance, uncertainty estimation, and
accuracy estimation for classification, e.g.,
with check models.

The proposed method on section 3 introduces
another model h to estimate the MSE that
seeks to minimize eq (11).
This section also suggests that optimizing eq (11) with
the training distribution instead of the
operational one is practically valid.
Section 3.2 establishes the Rademacher complexity
of models trained with this method,
and section 3.3 proposes a regularization.
The experiments in section 4.1 investigate regressing onto
the synthetic functions in equation (19)
and the experiments in section 4.2 look at some
LIBSVM datasets.

**Strengths:**

1. Understanding the generalization and downstream performance
   of a model is an important open topic and methods such
   as this have the potential to be impactful.
2. The experimental results quantified in Tables 1 and 3
   show that the proposed method best-optimizes for the
   generalization MSE compared against many relevant baselines

**Weaknesses:**

I am giving this paper a low-confidence review as I am not
an expert in this sub-area. I hope the following comments
as an outsider will be useful for clarifying the contribution.

1. The main weakness I see is in the experimental evaluation.
  The paper jointly proposes new settings for measuring
  the generalization MSE along with a new method.
  It is difficult to understand if the improved performance
  is because the method is better, or if the baselines
  are not well-tuned or faithfully reproduced.
  I would have found the experimental results significantly
  clearer if they used the exact experimental settings from
  the existing literature on related methods.
2. One questionable part about the experimental results
   is that their approach was tuned with a hyper-parameter
   search resulting in the hyper-parameters in Table B.1
   by selecting the hyper-parameters with the best MSE.
   Then, if I understand correctly, the baseline methods were
   evaluated using these hyper-parameters.
   If this is true, it is unfair to the baselines as they
   were never tuned for the MSE.
3. On the method, I find it empirically surprising that training another
   model h on the training set has any impact on
   the generalization performance of the model.
   The complexity bounds in section 3.2 explain this,
   but I still have a difficult time understanding how to
   interpret the complexity bounds and relating them
   to other approaches.
4. Despite section 2.1 presenting the training and
   operational distributions as potentially different,
   the theoretical results in section 3.2 and all of the
   experiments take the training and
   operational distributions to be the same.
5. The readability of the paper could be improved:
   + a) I found it confusing to read the problem setting in
      Section 2.1 as "preliminary" information even though
      it doesn't cite a reference for this problem setting.
      The related works presented afterwards in Section 2.2
      often do not look at exactly the formulation of Definition 1,
      so it would be good to specifically give a reference
      for other works solving exactly Definition 1.
   + b) This problem statement appears to extend
      the setting in [Chen 2021a](https://arxiv.org/pdf/2106.15728.pdf)
      from measuring the heldout accuracy of a classifier
      to the heldout MSE of a regression model.
      It would be good to clearly say and cite this again
      when defining the problem statement.
    + c) I found it somewhat confusing at first the paper uses
      the shorthand term "MSE" to refer to equation (1)
      which measures the MSE on an operational distribution.
      For example, this is not clear at all in the abstract,
      where I originally thought "MSE estimation" was referring
      to something else.

**Questions:**

I have given the paper a low-confidence review where the weaknesses
I see could be from my lack of understanding of this research sub-area.
I am extremely open to re-evaluating my score and discussing the points
I have brought up, especially on the experimental components.

---

### Author Response · Authors · 2023-11-22

We respectfully convey our gratitude to the reviewers for the abundant and valuable feedback, which will contribute to the constructive improvement of our paper's quality.
Regrettably, we have decided to withdraw our paper following a thorough assessment of the comments.
It has become apparent that the paper possesses shortcomings in theoretical analysis and experimental evaluation and requires a major revision.
Finally, we would like to once again thank the reviewers.